# Genomic Confluence: When Cerebrotendinous Xanthomatosis, Klinefelter Syndrome, and a BRCA2 Variant Intersect

**DOI:** 10.3390/ijms262110510

**Published:** 2025-10-29

**Authors:** Harry Pachajoa, Sebastián Bonilla, Daniel Andrés Nieva-Posso

**Affiliations:** 1Genomic Medicine Laboratory, Department of Basic Medical Sciences, Faculty of Health Sciences, Universidad Icesi, Cali 760031, Colombia; 2Departamento de Genética Clinica, Fundación Valle del Lili, Cali 760032, Colombia; sebonav17@gmail.com; 3Programa de Medicina, Universidad del Valle, Cali 760042, Colombia; nieva.daniel@correounivalle.edu.co

**Keywords:** xanthomatosis, cerebrotendinous, Klinefelter syndrome, BRCA2 protein, whole exome sequencing, genetic diseases, inborn, genetic counseling

## Abstract

Multilocus pathogenic variation—when multiple genetic disorders coexist in a single individual—remains rare but is increasingly recognized in the era of genomic medicine. Reporting such cases is essential for improving diagnostic accuracy, refining clinical management, and informing genetic counseling. We describe a pediatric case with a complex phenotype resulting from the coexistence of two distinct genetic diagnoses—cerebrotendinous xanthomatosis (CTX), a rare autosomal recessive lipid storage disorder caused by biallelic mutations in the *CYP27A1* gene and Klinefelter syndrome a common sex chromosome aneuploidy occurring in approximately 1 in 600 males, characterized by hypogonadism, gynecomastia, pubertal delay, infertility, micrognathia, and neurodevelopmental challenges—and an additional incidental finding with clinical relevance. The patient was born to consanguineous parents, presented with neurological symptoms, gastrointestinal dysfunction, endocrine abnormalities, and dysmorphic features. Trio-based exome sequencing identified a homozygous pathogenic variant in *CYP27A1* consistent with CTX, while conventional G-banded karyotyping revealed a 47,XXY chromosomal pattern, confirming Klinefelter syndrome. Additionally, a heterozygous pathogenic variant in *BRCA2* was incidentally detected, associated with hereditary cancer predisposition. The overlapping manifestations of CTX and Klinefelter syndrome produced a non-classical presentation that delayed diagnosis. Although the *BRCA2* variant did not contribute to the current phenotype, it has important implications for future cancer surveillance and family risk assessment. This case underscores the importance of combining classical cytogenetic and modern genomic methods to elucidate complex phenotypes, particularly in consanguineous populations, and highlights the need for the multidisciplinary management of patients with multilocus or incidental findings.

## 1. Introduction

Advancements in massive parallel sequencing technologies, particularly whole exome sequencing (WES), have significantly increased the detection of multiple independent genetic findings in a single individual—a phenomenon referred to as multigene or multilocus diagnoses [1]. Far from anecdotal, the estimated prevalence of such findings ranges from 4% to 7% in clinical genomic sequencing studies, with higher rates reported among patients with neurological disorders, consanguinity, atypical phenotypes, multisystem involvement, or protracted diagnostic journeys [2,3]. The identification of multiple pathogenic variants provides a more comprehensive framework for understanding complex clinical presentations, enhances diagnostic accuracy, and enables the implementation of personalized therapeutic and preventive strategies.

The recognition of dual or even triple genetic diagnoses is becoming increasingly relevant in clinical genetics, particularly when each variant carries distinct etiologic, prognostic, therapeutic, or familial implications [4]. This diagnostic complexity aligns with the principles of precision medicine, wherein integrated genomic data allow clinicians to manage patients not only according to the primary disease but also in consideration of coexisting genetic contributions that may modulate phenotypic expression, disease progression, or treatment response. Furthermore, acknowledging the coexistence of multiple genetic conditions can help avoid diagnostic anchoring, shorten the diagnostic timeline, and improve reproductive counseling and risk stratification for family members [5].

Cerebrotendinous xanthomatosis (CTX, MIM: 606530) is a rare autosomal recessive lipid storage disorder caused by biallelic mutations in the cytochrome P450 family 27 subfamily A member 1 (*CYP27A1*) gene, which encodes sterol 27-hydroxylase. Enzymatic deficiency leads to impaired bile acid synthesis and accumulation of cholestanol and cholesterol in the brain, tendons, and other tissues. CTX typically manifests in childhood with chronic diarrhea, juvenile cataracts, epilepsy, neuropsychiatric disturbances, hyporeflexia, and progressive cognitive decline [6,7].

*Klinefelter syndrome* (47,XXY) is a common sex chromosome aneuploidy occurring in approximately 1 in 600 males, and it presents with a broad phenotypic spectrum that may include hypogonadism, gynecomastia, pubertal delay, infertility, micrognathia, and neurodevelopmental challenges. Despite being a well-characterized condition, its clinical variability often contributes to delayed or missed diagnoses, particularly in pediatric populations [8,9].

Additionally, pathogenic variants in the breast cancer 2 susceptibility gene (*BRCA2)* are more commonly associated with hereditary breast and ovarian cancer syndromes in adults. However, such variants may be incidentally identified in pediatric patients undergoing WES for unrelated indications. The presence of a heterozygous *BRCA2* mutation confers increased lifetime risks for male breast cancer, prostate cancer, pancreatic cancer, melanoma, and central nervous system (CNS) tumors, thus warranting early cancer surveillance strategies and genetic counseling for the patient and at-risk relatives [10].

In this report, we present the case of a 14-year-old male with a complex constellation of neurological, gastrointestinal, endocrine, and dysmorphic features, in whom two clinically relevant and genetically distinct diagnoses were identified, CTX and Klinefelter syndrome, responsible for the clinical phenotype, as well as an incidental pathogenic variant in *BRCA2*. This case exemplifies the diagnostic and clinical value of combining conventional cytogenetic analysis with modern genomic sequencing to elucidate complex pediatric phenotypes and guide multidisciplinary management.

## 2. Case Description

A 14-year-old male patient, born in Venezuela and currently residing in Cali, Colombia, was referred to the medical genetics service due to early-onset seizures (most recently occurring two months prior to evaluation), mild cognitive impairment, gynecomastia, facial dysmorphisms, and intermittent chronic diarrhea. The family history was notable for parental consanguinity, as the patient’s parents are first cousins on the paternal side. Additionally, consanguinity was documented in the previous generation, with a consanguineous union among the patient’s great-grandparents The patient’s mother, a 40-year-old woman with no relevant medical history, reported an uneventful pregnancy and a term cesarean delivery due to fetal macrosomia.

Since the age of 3, the patient has experienced nocturnal focal motor seizures involving the right hemibody, which have been refractory to multiple antiepileptic drugs, including oxcarbazepine, valproic acid, and low doses of levetiracetam. Currently, he maintains 500 mg every 8 h. According to his mother, cognitive and language neurodevelopment was age-appropriate until approximately 6 years of age, at which point academic challenges emerged, accompanied by a progressive decline in attention, academic performance, and behavioral regulation. Neuropsychological assessment reported an Intelligence Quotient (IQ) of 70, evaluated with the Wechsler Intelligence Scale for Children (WISC), consistent with mild intellectual disability. A six-hour electroencephalogram telemetry showed bursts of generalized sharp waves, and brain Magnetic Resonance Imaging (MRI) revealed a subcortical malacic lesion in the right basal frontal region, interpreted as a chronic structural abnormality with potential epileptogenic significance. However, the MRI report did not include quantitative parameters such as lesion size or precise localization, which limits the descriptive value of the finding.

The mother also reported intermittent chronic diarrhea since the age of 4, with variable frequency, no seasonal pattern or dietary triggers, characterized by Bristol type 6 stools without mucus, blood, or steatorrhea. A gastroenterology consultation suggested a possible metabolic or genetic etiology. Basic metabolic screening revealed elevated total cholesterol (220 mg/dL) and low-density lipoprotein (LDL) cholesterol (140 mg/dL). Endocrine evaluation showed total testosterone 197 (ng/Dl), FSH 27.8 (mIU/mL), LH 18.5 (mIU/mL), consistent with the gonadal dysfunction typically observed in Klinefelter syndrome.

On physical examination, the patient measured 159 cm in height, weighed 54 kg, and had a head circumference of 57 cm, all within normal percentiles. However, several facial dysmorphisms were noted: synophrys, webbed neck, epicanthal folds, left palpebral ptosis, dorsal telangiectasias, asymmetric auricles, micrognathia, bifid uvula, a café-au-lait spot on the trunk, mild pectus excavatum, cavus foot, and bilateral gynecomastia. This combination of morphological and neurological findings raised suspicion of a multisystemic genetic disorder (Figure 1).

Given the phenotypic complexity, the prolonged diagnostic odyssey, and the history of parental consanguinity, both trio-based whole exome sequencing (WES) and G-banding karyotyping were performed. The karyotype revealed a 47,XXY chromosomal pattern. Thirty metaphases were analyzed, all showing an extra X chromosome with no structural chromosomal abnormalities, thereby ruling out mosaicism (Figure 2).

Whole exome sequencing (WES) was performed using the Illumina DNA Prep with Exome 2.5 Plus Enrichment kit (Illumina Inc., San Diego, CA, USA)**,** which incorporates genomic content from Twist Bioscience (South San Francisco, CA, USA) covering clinically relevant coding regions. Libraries were sequenced on an Illumina NextSeq 2000 platform (Illumina Inc., San Diego, CA, USA), following the manufacturer’s protocol. Sequencing reads were aligned to the GRCh38/hg38 human reference genome, and variant calling for germline variants was performed using DRAGEN Enrichment v4.2.7 (Illumina Inc., San Diego, CA, USA). Variants were filtered using standard quality metrics (call quality > 20, coverage > 10×, allele fraction > 15%). Annotation and prioritization were carried out using the Emedgene (Illumina Inc., San Diego, CA, USA) and Franklin (Genoox Ltd., Tel Aviv, Israel) platforms, referencing multiple clinical databases including ClinVar, HGMD, gnomAD, and OMIM. Molecular analysis identified a pathogenic homozygous variant in the *CYP27A1* gene (c.1183C>T; p.Arg395Cys), which was confirmed in heterozygosity in both parents.

The variant’s pathogenicity was supported by ACMG criteria (PM3, PP3, PM2, PM5, PS3, PP5 criteria). Although serum cholestanol levels were requested, the results were not yet available. However, the clinical presentation and genetic findings were considered diagnostic of cerebrotendinous xanthomatosis (CTX), a rare autosomal recessive lipid storage disorder characterized by abnormal cholestanol accumulation and progressive neurological, gastrointestinal, and systemic involvement.

Given the documented consanguinity and the autosomal recessive inheritance pattern of CTX, a three-generation pedigree was constructed, illustrating paternal consanguinity and the segregation pattern of the homozygous *CYP27A1* pathogenic variant (Figure 3).

Finally, a heterozygous pathogenic variant in *BRCA2* (c.4936_4939del; p.Glu1646Glnfs*23), inherited from his mother, was identified as an incidental finding. This frameshift variant is associated with increased lifetime risk for several malignancies, including male breast cancer, prostate cancer, and pancreatic cancer and highlights the need for structured genetic counseling and long-term surveillance strategies for the proband and at-risk family members.

Following the diagnosis, the patient was referred for a multidisciplinary medical board evaluation to determine the initiation of treatment. He is currently awaiting this assessment.

## 3. Discussion and Conclusions

This case illustrates the diagnostic and clinical challenges associated with multilocus pathogenic variation in pediatric patients with complex, multisystemic presentations. The simultaneous identification of cerebrotendinous xanthomatosis (CTX) and Klinefelter syndrome (47,XXY), together with an incidental *BRCA2* pathogenic variant, underscores the increasing relevance of considering multilocus diagnoses in individuals with consanguinity, atypical evolution, or phenotypic expansion [11].

The systematic implementation of high-throughput sequencing technologies, such as whole exome sequencing (WES), has shifted the diagnostic paradigm by enabling the detection of multiple causative variants in a single individual. Current studies estimate that 4% to 7% of patients undergoing genomic testing harbor more than one molecular diagnosis, which may result in blended phenotypes or overlapping phenotypes [11]. This phenomenon, referred to as “genetic collision”, requires a comprehensive, multidisciplinary diagnostic framework.

Comparable reports in the literature support this emerging complexity. Posey et al. [12], found that 31.6% of patients with presumed phenotypic expansion had multilocus pathogenic variation when reassessed by exome sequencing. Other published cases have identified up to four concurrent diagnoses in a single patient [5], while dual molecular diagnoses—including rare combinations such as Angelman syndrome with Krabbe disease—have been reported in children with atypical disease progression [13]. These findings underscore the growing need to adopt multigenic models in clinical reasoning.

For the patient in this report, clinical management required a domain-specific and integrated approach. CTX is a treatable metabolic disorder; the administration of chenodeoxycholic acid can restore bile acid homeostasis and prevent the progression of neurological symptoms [14]. Klinefelter syndrome management includes testosterone replacement therapy to support pubertal development, improve bone density, and address neurobehavioral outcomes [15]. The *BRCA2* variant, although not responsible for the presenting phenotype, has actionable implications and necessitates long-term cancer surveillance and cascade testing for relatives, accordance with National Comprehensive Cancer Network (NCCN) guidelines [16].

A detailed comparison of the patient’s clinical findings with the classic manifestations of each disorder revealed overlapping neurological and endocrine features (Table 1). While CTX primarily accounted for the neurological and gastrointestinal symptoms, and Klinefelter syndrome explained the endocrine and some dysmorphic traits, the co-occurrence of both contributed to a blended, non-classical presentation that delayed diagnosis. In contrast, the *BRCA2* variant did not contribute to the observed phenotype but represents an incidental, clinically actionable finding with implications for long-term cancer surveillance and family counseling. Although no formal quantitative scoring system was applied, the contribution of each condition to the overall phenotype was qualitatively analyzed based on disease expressivity and published clinical spectrum. Moreover, it is conceivable that the endocrine disturbances and altered lipid metabolism associated with Klinefelter syndrome may exacerbate the metabolic imbalance characteristic of CTX, although this interaction remains speculative and requires further study [6,8].

The concurrent identification of this combination is striking. Assuming statistical independence, the joint prevalence of CTX (0.0003%) [17], Klinefelter syndrome (0.17%) [18], and *BRCA2* pathogenic variants (0.25%) [19] yields an estimated probability of approximately 1.275 × 10^−11^—or roughly one in 78 billion births—emphasizing the uniqueness of this case.

Importantly, the *BRCA2* variant was detected as an incidental finding. However, this variant (c.4936_4939del; p.Glu1646Glnfs*23) is a well-established pathogenic mutation that is included in current cancer screening recommendations. Its disclosure is aligned with current American College of Medical Genetics and Genomics (ACMG) guidelines, which, in version 3.2 of their secondary findings list, include *BRCA2* among 81 genes whose pathogenic variants must be reported due to their actionable nature and potential to significantly reduce morbidity through early intervention [20].

Beyond its clinical novelty, this case underscores the value of combining traditional cytogenetic methods with modern genomic sequencing to achieve a complete diagnosis. Integrating both approaches facilitated the recognition of a dual etiologic diagnosis and an additional actionable secondary finding. Furthermore, this case highlights the importance of establishing protocols for individuals with multilocus diagnoses. These may include integrated follow-up frameworks, criteria for therapeutic prioritization when multiple conditions co-occur, and expansion of genetic counseling models to address both treatable and predictive conditions simultaneously. As the field of genomic medicine evolves, there is a growing need to incorporate multigenic reasoning into clinical training and decision-making pathways [13,21].

This case exemplifies the increasing diagnostic and clinical complexity introduced by multilocus genetic findings in pediatric medicine. The co-occurrence of a treatable metabolic disorder and a syndromic aneuploidy in the same patient underscores the importance of an integrated, multidisciplinary approach to care. The case highlights the clinical utility of WES, cytogenetic analysis as complementary tools for precision diagnosis, comprehensive management, and ethically guided handling of incidental findings. Finally, it reinforces the urgent need to develop evidence-based guidelines for the diagnosis, genetic counseling, and long-term care of patients with multiple molecular diagnoses.

## Figures and Tables

**Figure 1 ijms-26-10510-f001:**
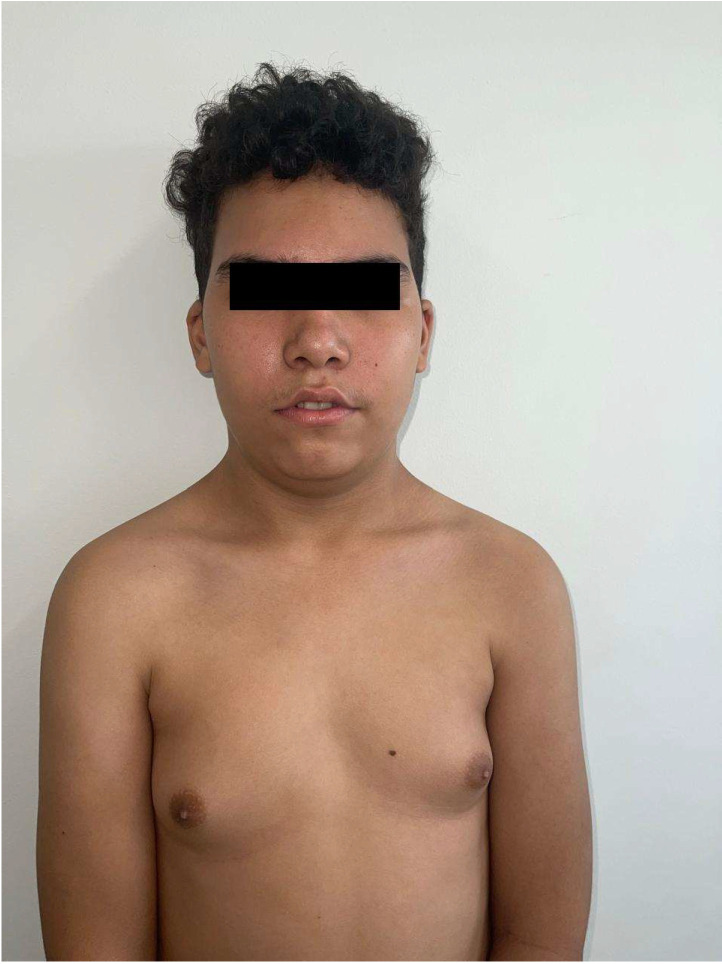
Dysmorphic features and physical findings observed in the patient.

**Figure 2 ijms-26-10510-f002:**
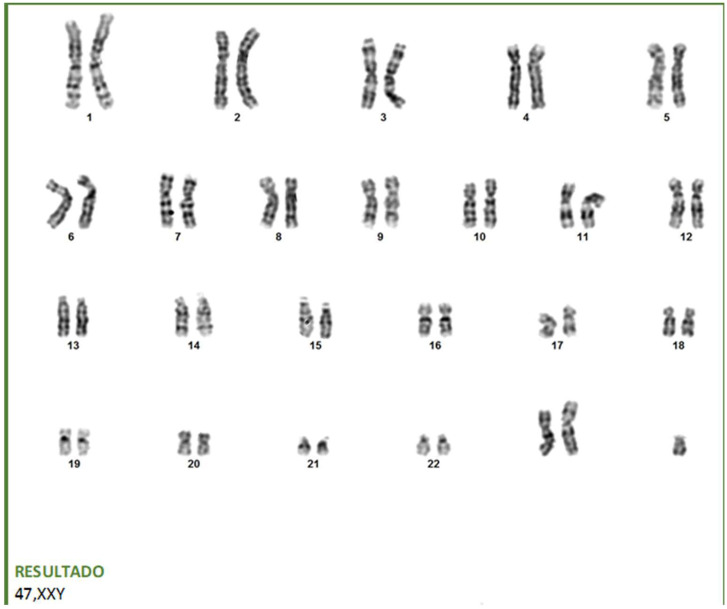
The Karyotype of the patient shows a 47,XXY chromosomal pattern.

**Figure 3 ijms-26-10510-f003:**
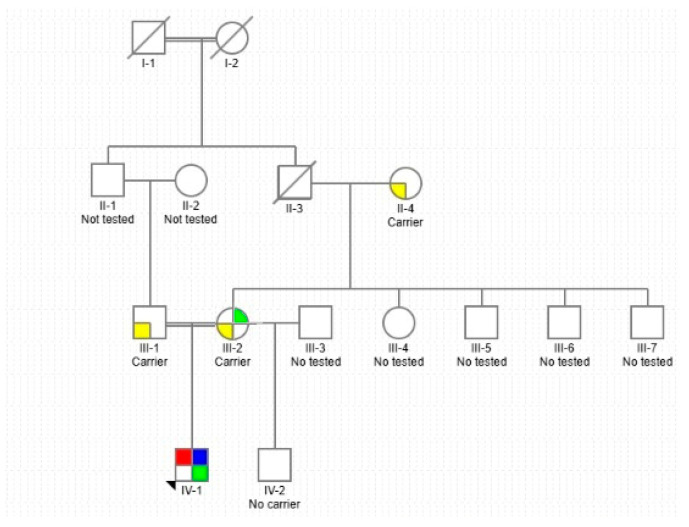
Pedigree showing segregation of genetic variants across four generations. Squares represent males, circles females, and slashed symbols indicate deceased individuals. The proband (IV-1), marked by an arrow, is homozygous for a pathogenic *CYP27A1* variant (red) and also carries variants associated with Klinefelter syndrome (blue) and *BRCA2* (green). Yellow indicates heterozygous carrier status. Individuals labeled “Carrier” are confirmed carriers; “No carrier” tested negative; “Not tested” indicates no genetic testing performed.

**Table 1 ijms-26-10510-t001:** A comparative analysis of the patient’s clinical features and the classic manifestations of CTX, Klinefelter syndrome, and BRCA2 variant.

	CTX	Klinefelter Syndrome	*BRCA2* Variant	Observed in Patient	Interpretation/Likely Contribution
Neurological					
Seizures	+	Rare	−	+	Explained by CTX (neurological involvement)
Cognitive impairment	+	+	−	+	Blended effect of CTX and Klinefelter
Behavioral difficulties/inattention	+	+	−	+	Likely additive contribution
MRI brain lesion	+	−	−	+	Consistent with CTX
Endocrine/Metabolic					
Gynecomastia	−	+	−	+	Typical for Klinefelter syndrome
Hypogonadism	−	+	−	+	Klinefelter-related endocrine dysfunction
Elevated cholesterol/LDL	+	−	−	+	Typical of CTX lipid metabolism
Growth parameters (normal)	Variable	Often tall stature	−	+	Compatible with Klinefelter
Gastrointestinal					
Chronic diarrhea	+	−	−	+	Characteristics of CTX
Dysmorphic/Physical					
Micrognathia	Rare	+	−	+	Likely Klinefelter-associated
Synophrys/ptosis/epicanthal folds	Variable	Mild	−	+	Possibly overlapping or nonspecific
Webbed neck/pectus excavatum	Rare	Occasional	−	+	Possibly related to Klinefelter
Café-au-lait spot/telangiectasias	−	−	−	+	Incidental, unrelated
Genetic Findings					
*CYP27A1* homozygous variant (c.1183C>T)	Diagnostic	−	−	+	Confirms CTX
*BRCA2* heterozygous variant (c.4936_4939del)	−	−	Diagnostic	+	Incidental, cancer predisposition
47,XXY karyotype	−	Diagnostic	−	+	Confirms Klinefelter

+ indicates the presence of the feature; − indicates its absence.

## Data Availability

Not applicable.

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
