# Peer review of "Genomic Confluence: When Cerebrotendinous Xanthomatosis, Klinefelter Syndrome, and a BRCA2 Variant Intersect"

_ijms, 2025, doi:10.3390/ijms262110510_

Round 1
Reviewer 1 Report
Comments and Suggestions for Authors
The article "Genomic Confluence: When Cerebrotendinous Xanthomatosis, Klinefelter Syndrome, and BRCA2 Variant Intersect" describes a clinical case involving a combination of three conditions (according to the authors). However, it would be more accurate to describe this as a combination of only two conditions, as only the symptoms of cerebrotendinous xanthomatosis and Klinefelter syndrome lead to the manifestation of atypical symptoms. The third finding, a BRCA2 variant, is an incidental finding that does not significantly affect the clinical presentation at this time. The article could still mention this finding, but it should be described as an incidental observation rather than a third condition.
In the abstract, the authors write that all anomalies were identified using exomic sequencing. However, in the text of the article, it is stated that Klinefelter syndrome was detected using G-banding.
The authors begin line 67 by introducing the word "In contrast", but it is not clear exactly what the contrast refers to.
In the description of the karyotyping results, the karyotype is indicated as 46, XXY, with a space in between. According to the International System for Human Cytogenetic Nomenclature (ISCN), there should be no space. It would also be helpful to include information about the number of cells analyzed and the number of abnormal cells (if mosaicism was present), in accordance with the ISCN guidelines.
It would be more appropriate to move the description of the BRCA2 mutation to the very end of the Results section, after all other results have been presented.
In the Discussion and Conclusions section, the authors discuss the shift in diagnostic paradigm with the introduction of WES. At the same time, they describe the findings of their study, which were obtained through a combination of modern and classical methods. Perhaps the authors should consider the benefits of combining traditional and modern diagnostic methods.
Author Response
Dear Reviewer,
We appreciate the constructive feedback provide during the review process. We have revised the text accordingly and describe below the specific changes made in response to each point:
- “The article should describe a combination of only two conditions; BRCA2 is an incidental finding.”
We agree with this observation. The text has been revised to describe two clinically relevant diagnoses (CTX and Klinefelter syndrome) and the BRCA2 variant as an incidental, actionable finding without current phenotypic contribution.
- “In the abstract, it is stated that all anomalies were identified using exomic sequencing, but the text says that Klinefelter syndrome was detected using G-banding.”
The inconsistency has been corrected. The Abstract now specifies the diagnostic method for each finding.
- “The word ‘In contrast’ (line 67) does not clearly indicate what the contrast refers to.”
We have revised the sentence to improve clarity by replacing “In contrast” with “Additionally”, which more accurately conveys the intended continuation of ideas rather than opposition
- “The karyotype is written as ‘47, XXY’ with a space; according to ISCN, there should be no space. It would also be helpful to include the number of cells analyzed and whether mosaicism was present.”
Corrected to ISCN-compliant format and additional methodological details have been included.
- “It would be more appropriate to move the description of the BRCA2 mutation to the end of the Results section.”
We agree with the observation. The section has been reorganized so that CTX and Klinefelter syndrome are presented first, and the BRCA2 variant is discussed at the end of the Results section as an incidental finding.
- “In the Discussion and Conclusions, consider emphasizing the benefits of combining traditional and modern diagnostic methods.”
The Discussion has been expanded to explicitly highlight the complementary value of integrating classical cytogenetics (G-banding) with modern genomic approaches (WES) in complex cases.
Thank you once again for your thoughtful feedback and consideration.
Best regards,
The Authors
Reviewer 2 Report
Comments and Suggestions for Authors
This article presents a clinical case describing a unique combination of three genetic pathologies in a single patient: cerebrotendinous xanthomatosis (CTX), Klinefelter syndrome (47,XXY), and a pathogenic variant of the BRCA2 gene. This case illustrates the increasing diagnostic complexity in the era of genomic medicine, when several independent Mendelian conditions (multilocus pathogenic variation) can be identified in a single individual. The patient, a child of a consanguineous marriage, had a complex phenotype, including neurological symptoms (refractory epilepsy, cognitive deficit), gastrointestinal disorders (chronic diarrhea), endocrine abnormalities (gynecomastia), and dysmorphic features. The diagnosis was established using karyotyping and whole exome sequencing of the trio. The combined effects of CTX and Klinefelter syndrome resulted in an atypical clinical presentation, delaying diagnosis. Identification of the BRCA2 variant, although unrelated to the current symptoms, is critical for long-term cancer monitoring and family risk assessment. The authors emphasize the utility of whole-exome sequencing for unraveling complex phenotypes, especially in inbred populations, and the need for a multidisciplinary approach, personalized care for patients with multiple genetic diagnoses, and the importance of managing incidental findings.
However, it would be necessary to clarify a number of comments that are available to the article:
- A detailed comparison of the observed combined phenotype with the classic manifestations of each of the three diseases individually was not provided. A table systematizing the contribution of each diagnosis to specific symptoms (neurological, endocrine, dysmorphic) would significantly strengthen the interpretation.
- The Methods section lacks technical details of the whole exome sequencing (WES) performed: indication of the sequencing platform (e.g., Illumina), coverage depth, percentage of target regions covered at a given depth (e.g., >20x), and variation and validation methods (e.g., Sanger).
- For the CYP27A1 variant (c.1183C>T; p.Arg395Cys), it would be useful to mention whether functional in vitro studies were performed to confirm its pathogenic effect on enzymatic activity, or whether the conclusion is based solely on bioinformatic predictions and population frequency data.
- To assess the contribution of each genetic disorder to the overall phenotype, scoring methods could be used, or a prognostic model could be constructed that takes into account the penetrance and expressivity of each disease.
- When describing the MRI results ("subcortical malacic zone"), quantitative parameters (lesion size, precise localization by coordinates) are not provided, which reduces the information content.
- The number of metaphases analyzed and the G-banding resolution (number of bands per haploid genome), which is the standard for karyotyping reports, are not specified.
- To confirm the diagnosis of CTX, measurement of serum cholestanol levels or other biomarkers is critical.
- The assertion that the combination of CTX and Klinefelter syndrome led to a non-classical presentation is not supported by an analysis of potential pathophysiological interactions (e.g., the effect of hypogonadism on the neurological symptoms of CTX).
- There is a lack of data on the dynamics of the patient’s condition against the background of the initiated therapy (chenodeoxycholic acid and testosterone), which could be a valuable addition.
Author Response
Dear Reviewer,
We appreciate the constructive feedback provide during the review process. We have revised the text accordingly and describe below the specific changes made in response to each point:
- “A detailed comparison of the observed combined phenotype with the classic manifestations of each of the three diseases individually was not provided. A table systematizing the contribution of each diagnosis to specific symptoms would strengthen the interpretation.”
We appreciate this insightful suggestion. We have added Table 1, which systematically compares the patient’s clinical features with the classic manifestations of CTX, Klinefelter syndrome, and the BRCA2 variant, and interprets the likely contribution of each condition to the observed phenotype.
This addition strengthens the discussion by clarifying how overlapping and independent features were analyzed and attributed.
- “The Methods section lacks technical details of the whole-exome sequencing (WES).”
We have expanded the Methods subsection in the Case Description to include comprehensive WES parameters.
- “For the CYP27A1 variant (c.1183C>T; p.Arg395Cys), it would be useful to mention whether functional in vitro studies were performed.”
Thank you for the suggestion. We clarified that no in vitro functional assays were performed, and that the pathogenicity assessment was based on ACMG criteria, clinical correlation, and previously reported cases.
- “To assess the contribution of each genetic disorder to the overall phenotype, scoring methods could be used, or a prognostic model could be constructed.”
We appreciate this valuable recommendation. While quantitative scoring models are not yet validated for multilocus phenotypes, we conducted a qualitative comparative analysis, now summarized in Table 1, which describes the relative contribution of each condition based on known disease expressivity and the literature.
- “When describing the MRI results (‘subcortical malacic zone’), quantitative parameters are not provided.”
We agree. Unfortunately, the original MRI report did not include quantitative measurements such as lesion size or exact coordinates. We have clarified this limitation in the revised text to accurately reflect the available information.
- “The number of metaphases analyzed and the G-banding resolution are not specified.”
We have added this methodological information in accordance with ISCN recommendations.
- “To confirm the diagnosis of CTX, measurement of serum cholestanol levels or other biomarkers is critical.”
We agree with the reviewer and clarified this point. Serum cholestanol testing was requested but results were not yet available at the time of reporting; this is now explicitly stated.
- “The assertion that the combination of CTX and Klinefelter syndrome led to a non-classical presentation is not supported by analysis of potential pathophysiological interactions.”
We expanded this discussion to include a pathophysiological hypothesis based on the literature. Specifically, we propose that endocrine dysfunction and altered lipid metabolism in Klinefelter syndrome may exacerbate the metabolic imbalance in CTX, possibly influencing neurological progression.
- “There is a lack of data on the dynamics of the patient’s condition after initiation of therapy.”
We agree. At the time of writing, treatment had not yet been initiated. As clarified in the manuscript, the patient was referred for a multidisciplinary medical board evaluation to determine treatment initiation and is currently awaiting this assessment. Therefore, longitudinal post-treatment data are not yet available.
Thank you once again for your thoughtful feedback and consideration.
Best regards,
The Authors
Round 2
Reviewer 1 Report
Comments and Suggestions for Authors
Thank you for taking into account my comments and observations. Based on the corrections and additions that you have made (apparently in response to the comments of another reviewer), I do not have any further questions or comments regarding the article.
Author Response
Dear Reviewer,
We sincerely thank you for your careful evaluation of our manuscript and your constructive feedback. We are pleased that the revised version of the manuscript addresses all previous concerns.
Best regards,
The Authors
Reviewer 2 Report
Comments and Suggestions for Authors
This article presents an exceptional clinical case of coexpression of three genetically independent conditions in a single patient. The relevance of this work is consistent with current trends in clinical genetics, where multilocus diagnoses are no longer a casuistry. The authors have significantly revised the manuscript, adequately addressing most of the comments.
- Regarding comparative phenotype analysis: The authors fully implemented the recommendation by adding Table 1, which has become a key element of the article. The table systematizes the contribution of each disease to specific symptoms and significantly enhances the interpretation of a complex phenotype. It is particularly valuable that the authors not only listed the symptoms but also provided a substantiated interpretation of their origins ("explained by CTX," "mixed effect," "additive contribution"). 2. Regarding WES methods: The "Methods" section has been supplemented with technical details (use of Illumina DNA Prep with Exome 2.5 Plus Enrichment, NextSeq 2000 platform, analysis parameters), which is consistent with current standards for describing sequencing methods.
- Regarding functional studies: The authors correctly stated that functional in vitro studies were not conducted, and the pathogenicity assessment is based on ACMG criteria and clinical correlation. For the identified pathogenic variant in CYP27A1, this is sufficient.
- Regarding disease contribution assessment: The qualitative comparative analysis in Table 1 adequately replaces quantitative assessment methods, which are indeed not validated for multilocus phenotypes. The authors correctly limited themselves to qualitative analysis based on known disease expressivity.
- Regarding MRI data: Honestly acknowledging the lack of quantitative parameters in the original MRI report is the correct approach. This reflects real-world clinical practice.
- Regarding cytogenetic methods: Adding information on the number of metaphases analyzed (30) complies with ISCN standards and improves the reliability of karyotyping.
- Regarding CTX biomarkers: Clarifying that the serum cholestanol level was requested but the results are unavailable eliminates this issue. For genetically confirmed CTX, this is not a critical deficiency.
- Regarding pathophysiological interactions: The authors significantly strengthened the discussion by putting forward a reasonable hypothesis about the possible mutual influence of endocrine dysfunction in Klinefelter syndrome and metabolic disturbances in CTX. This is an important conceptual addition.
- Regarding treatment data: Clearly stating that treatment has not yet been initiated and the patient is awaiting a multidisciplinary assessment eliminates concerns about the lack of follow-up data.
Despite significant improvements to the manuscript, several issues remain requiring attention:
- Clinical details: an IQ of 70 is listed, but the test used (WISC, Stanford-Binet, etc.) is not specified; antiepileptic therapy, including drug doses and treatment duration, is incompletely described; hormonal status data (testosterone, LH, FSH), which are important for Klinefelter syndrome, are missing.
- Discussion of BRCA2: It should be clarified that the BRCA2 variant (c.4936_4939del; p.Glu1646Glnfs*23) is an established pathogenic variant and is included in screening recommendations, even though it is an incidentaloma.
Author Response
Dear Reviewer,
We thank the reviewer for their thorough and insightful comments. All the suggested corrections have been addressed as detailed below.
- “An IQ of 70 is listed, but the test used (WISC, Stanford-Binet, etc.) is not specified.”
We agree. This information has been added. The text now specifies that the IQ was evaluated using the Wechsler Intelligence Scale for Children (WISC), consistent with mild intellectual disability.
- “Antiepileptic therapy, including drug doses and treatment duration, is incompletely described.”
We have expanded this section to include details of antiepileptic treatment history and current management.
- “Hormonal status data (testosterone, LH, FSH), which are important for Klinefelter syndrome, are missing.”
We appreciate this insightful suggestion. Hormonal evaluation results have been incorporated as requested.
- “It should be clarified that the BRCA2 variant (c.4936_4939del; p.Glu1646Glnfs23) is an established pathogenic variant and is included in screening recommendations, even though it is an incidentaloma.”
We agree with this observation. We revised the discussion to include this clarification.
Thank you once again for your thoughtful feedback and consideration.
Best regards,
The Authors